# Is There a Futility Discriminant Function Score for Alcoholic Hepatitis?

**DOI:** 10.3390/jcm10132756

**Published:** 2021-06-23

**Authors:** Kevin Lamm, Maggie McCarter, Mark W. Russo

**Affiliations:** 1Carolinas Medical Center, Department of Internal Medicine, Division of Hepatology, 6th Floor Morehead Medical Plaza, 1025 Morehead Medical Drive, Charlotte, NC 28204, USA; kevin.lamm@atriumhealth.org; 2Carolinas Medical Center, Center for Outcomes Research and Evaluation, Research Office Building, 1000 Blythe Blvd Charlotte, NC 28204, USA; maggie.mccarter@atriumhealth.org

**Keywords:** alcoholic hepatitis, discriminant function, futility, mortality

## Abstract

The treatment for patients with alcoholic hepatitis (AH) who have a discriminant function (DF) score greater than 32 has been steroids. A prior study reported that mortality approaches 100% when the DF score is greater than 54, despite the use of prednisolone. Our aim was to determine if a DF score greater than 54 is associated with high mortality despite steroids. We conducted a retrospective study of 531 first-time inpatient encounters of AH. We compared 30-day mortality between patients with DF 54 or less to those greater than 54 treated with steroids, as well as a matched group not treated with steroids. A total of 531 inpatients diagnosed with AH were identified, of which 124 had a DF greater than 32 and 52 were treated with steroids. Among patients treated with steroids, 30-day mortality for patients with DF greater than 54 (*n* = 27) and 54 or below (*n* = 25) was 22% and 12%, respectively (*p* = 0.47). Among patients with DF greater than 54, the 30-day mortality for those who did (*n* = 27) and did not (*n* = 29) receive steroids was 22% and 24%, respectively (*p* = 0.87). In our study population, steroids were not futile in patients with a DF score of greater than 54.

## 1. Introduction

Alcohol use disorder and high-risk drinking rates increased by 36% and 49%, respectively, over a 10-year span [1]. The National Institute on Alcohol Abuse and Alcoholism analyzed data from 2015 and determined that within the previous month, 56% of adults reported drinking at least one alcoholic beverage and 27% admitted to binge drinking [1]. The degree of injury to the liver from alcohol is not linearly dose-dependent, but there appears to be a threshold above which the ongoing consumption of alcohol causes liver injury [2]. For men, the threshold for harmful drinking is four alcohol drinks per day and for women it is two alcohol drinks per day. Binge drinking is defined as men who drink five or more drinks or women who consume four or more drinks in 2 h.

The incidence of alcoholic hepatitis (AH) has been difficult to accurately estimate because its diagnostic coding is not reliable [3]. Pang et al. analyzed a four-and-a-half-year period of hospitalized patients by applying ICD 9 or 10 diagnostic codes and lab criteria. They determined that using ICD codes with laboratory studies was superior to ICD coding alone, with a positive predictive value of 54%. The negative predictive value could not be calculated due to a lack of an unaffected control group [3]. However, the AH consortia validated a protocol using coding and labs that accurately identifies patients with AH [4].

Since Maddrey, et al.’s landmark paper in 1978, patients with a DF score >32 have been considered for treatment with corticosteroids [5]. The STOPAH trial evaluated the effect of treatment and found that therapy with prednisolone resulted in a statistically significant decrease in 28-day mortality but did not improve mortality at 90 days or 1 year [6]. A meta-analysis performed by Louvet et al. found that steroids when compared with pentoxifylline or placebo showed a decrease in mortality with steroids at 28 days but not at 6 months [7]. Infection has typically been a contraindication to giving steroids for AH but a meta-analysis by Hmoud et al. suggested a 28-day improvement in mortality in patients with steroids and no difference in mortality based on whether or not patients were treated with steroids [8]. The most recent American Association for the Study of Liver Disease guideline for alcoholic hepatitis does not provide a DF at which treating with steroids is contraindicated [9]. A Veterans Affairs study in 1995 reported one- and six-month mortality based on the DF score, which showed that in the range of 35–54, mortality was reduced with steroids, but those with a DF score >54 steroids did not obtain benefits and these scores were associated with lower survival, suggesting a threshold above which corticosteroids are harmful [10]. Twenty-five years have elapsed since this study and significant advances have been made in treating advanced liver disease. The aim of this study was to determine if the use of steroids was futile in patients with an AH and DF greater than 54.

## 2. Patients and Methods

### 2.1. Patient Selection

We identified AH patients who were hospitalized during January 2010–December 2018, using the Atrium Health Electronic Medical Record. The Atrium Health Electronic Medical Record includes information from all hospitals within the healthcare system including Carolinas Medical Center and 11 regional hospitals, with a combined total of 3000 beds. We excluded patients less than age 18.

### 2.2. Definition of AH

To determine the diagnosis of AH, we applied the AH consortia methodology [4]. We screened patients with the ICD 9 and ICD 10 codes (571.1 and K70.1), including those that that had obtained a primary or secondary diagnosis from Atrium Health’s Electronic Medical Record. A total of 531 patients were identified. To increase the study’s specificity, only patients with the following lab values were included in the study: AST > 50 U/L, AST < 400 U/L, ALT < 400 U/L, T. Bili > 3 mg/dL, and a AST/ALT ratio of >1.5, as recommended by the AH consortia [4]. Afterwards, 260 records were utilized after screening with the inclusion criteria (Figure 1).

### 2.3. Data Collection

Data was collected on patient demographics as well as 30- and 90-day mortality. Laboratory values analyzed included white blood cell count, platelets, sodium, creatinine, prothrombin time, INR, bilirubin day 0 and 7, albumin, ALT, and AST. The prognostic scores were calculated for all patients using the first lab taken after inpatient admission. Afterwards, 30- and 90-day mortality were gathered by using the social security database. Presence of infection or GI hemorrhage was defined by ICD 9 and ICD 10 codes (see Appendix A). The Charlson Comorbidity Index was calculated using the Atrium Healthcare ICD 9 and 10 coding algorithms. All data entry and storage were kept secure in an Excel spreadsheet.

### 2.4. Objective of the Study

The primary endpoint of the study was 30-day mortality. The secondary endpoint was 90-day mortality. The objective of the study was to determine if there is a DF score at which giving steroids is futile.

### 2.5. Statistical Analysis

Using descriptive statistics and univariate analyses, baseline characteristics were compared between the 33–54 and >54 DF score groups in patients who received steroids, and between steroid- and no-steroid treatment groups in patients with a DF greater than 54. Continuous variables are reported as medians and interquartile ranges, and categorical variables are reported as frequencies and percentages. Between-group comparisons for baseline characteristics were assessed using Student’s *t*-test for normally distributed variables, non-parametric Kruskal Wallis tests for continuous variables that were not normally distributed, and Pearson’s chi-square test or Fisher’s exact tests for categorical variables. The groups were compared on 30-day and 90-day mortality using the chi-square test or Fisher’s exact test. Primary analyses of the relationship between DF, Creatinine, Albumin, and steroid treatment, and 30- and 90-day mortality among those with a DF >32 were performed using multiple logistic regression models adjusting for GI bleeding, age, and the Charlson Comorbidity Index. Secondary analyses of the relationship between DF and 30- and 90-day mortality among those treated with steroids were performed using multiple logistic regression models adjusting for age and creatinine lab value, and the Charlson Comorbidity Index and creatinine separately. All statistical tests were two-sided with a significance level of 0.05, and were conducted using SAS/STAT, Version 9.4 (SAS Institute Inc., Cary, NC, USA).

## 3. Results

### 3.1. Study Population

From January 2010 through December 2018, 531 patients were given a primary or secondary discharge diagnosis of AH. Of those, 260 met the laboratory inclusion criteria. In addition, 124 of those patients had DF > 32 with 56 and 68 patients had DF > 54 and DF 54 or less, respectively. Baseline characteristics of the 52 patients who received steroids with DF greater than 32 are shown in Table 1, stratified by a DF score of 33–54 and greater than 54. Variables that were significantly different between groups include serum sodium, creatinine, PT/INR, albumin, and diagnosis of infection (defined in Appendix A). Baseline characteristics of patients with DF greater than 54 are summarized in Table 2 and compared based on whether or not they received steroids. Patients who received steroids had a significantly lower serum sodium, higher serum AST, and a higher percentage of diagnosed infection.

### 3.2. Patient Survival

Figure 2 compares 30-day and 90-day mortality based on the DF score (≤54, >54) and steroid treatment (yes/no). For patients with DF >54 who did (*n* = 27) and did not (*n* = 29) receive steroids, 30-day mortality was 22% and 24%, respectively (*p* = 0.87). For those with DF ≤54 but >32, 30-day mortality for those who did (*n* = 25) and did not (*n* = 43) receive steroids was 12% and 16%, respectively (*p* = 0.66).

In an additional effort to find a DF score that would suggest futility, we compared outcomes in patients with a DF score > 65. For patients with DF > 65 who did (*n* = 15) and did not (*n* = 17) receive steroids, 30-day mortality was 20% and 29%, respectively (*p* = 0.69). Table 3 shows results of the multivariable logistic regression analyses for 30- and 90-day mortality in patients with DF > 32. Creatinine on admission was found to have a statistically significant impact on 30-day mortality, but not on 90-day mortality. Based on the International Club of Ascites defining acute kidney injury as an absolute increase in serum Cr of 0.3 mg/dL or more from baseline, we chose to look at whether an increase of 0.3 mg/dL of creatinine led to increased mortality [11]. For every 0.3-unit increase in creatinine, the odds of 30-day mortality increased by 19% (OR 1.19 95%CI 1.04–1.37). No other statistically significant associations were found.

Results of the logistic regressions performed on the relationship between 30- and 90-day mortality and DF within patients treated with steroids can be found in Table 4. There were no significant associations between DF and 30- and 90-day mortality.

## 4. Discussion

Alcoholic hepatitis is a common consult for gastroenterologists and for more than four decades, the decision of whether to initiate steroids has been a large part of its management. The controversy over steroids persists and whether steroids offer a survival benefit or do more harm than good continues to be discussed. Studies have demonstrated short term survival, or 30-day survival can be improved, but may come at the expense of increased infections, and long-term survival as 90-day or 6-month survival is not improved with steroids. Whether there is a point of futility or a DF score where severity of AH is so severe that steroids will not improve outcomes is not well described.

The DF score is applied as a dichotomous variable whereby individuals with a DF score greater than 32 are considered for steroids and those with scores 32 or less receive supportive therapy alone. However, a patient with a DF score of 55 may be much sicker than one with a score of 35. In a VA study that included 536 patients with AH and compared placebo, prednisolone, and oxandrolone, mortality was significantly lower in patients treated with prednisolone with DF between 35 and 54 compared to the placebo group. In patients with DF greater than 54, mortality was higher in the prednisolone group and approached 100% compared to a mortality of 75% in the placebo group, but it was not reported to be statistically significant [10]. In our study, in patients with DF > 54 mortality was 21% and 23% in patients treated with steroids and those not treated with steroids. Although our study suggests steroids do not improve survival in patients with DF > 54, larger prospective studies are needed to confirm this finding before definitive recommendations could be made to withhold steroids from patients with AH and DF scores > 54.

Since the development of the DF score, a number of other prognostic scores have been developed, but none have replaced the DF. A Glasgow Alcoholic Hepatitis Score of 9 or higher or a MELD score of 20 or higher have been proposed to select patients for steroids or liver-specific treatment for AH, but neither are substantially better at predicting mortality compared to the DF score [12,13]. The Lille Score at baseline and day 4 or 7 allows the clinician to assess the response to steroids. Non-responders can stop steroids and alternative therapies such as early liver transplantation should be considered [14]. For the most part, the DF score has stood the test of time and remains in guidelines for identifying patients with AH who may benefit from steroids [9].

The original study that developed DF reported improved survival in patients with AH with scores greater than 32. Patients were excluded if they had active gastrointestinal bleeding, pancreatitis, a history of peptic ulcer disease, active infection, or a history of viral hepatitis. The 55 subjects included in the study were divided into three groups based on severity of illness defined by presence of ascites, encephalopathy, coagulopathy, and bilirubin. Higher mortality was associated with encephalopathy, coagulopathy, hypoalbuminemia, higher bilirubin, higher BUN, and creatinine. All deaths occurred in patients with a DF score of 93 or higher; 6 of 8 placebo patients died and 1 or 7 prednisolone patients died (*p* = 0.03) [5].

The limitations of our study include those associated with retrospective studies and the biases associated with selecting a control group. Patients with DF > 54 who were selected to receive steroids may have been systematically different than patients with DF > 54 who did not receive steroids. Various confounders were controlled for to address this bias. Creatinine is not included in the DF score, but it is included in the MELD score and the effect of renal injury on futility of steroids in AH is worth further study with MELD. Cases were identified using ICD codes; however, we used rigorous methods previously identified by the Alcohol Research Consortium that have been validated and have excellent accuracy for identifying patients with AH [3,4]. Although our study included a large number of overall patients with AH, the number of patients in subgroups was smaller and we may have been underpowered to detect a clinically significant difference between groups with high and low DF scores.

In conclusion, in contrast to the VA study, we did not find that patients with AH and DF scores greater than 54 had higher mortality compared to those with lower DF scores. Even in patients with severe AH, steroids do not appear to be futile. We advocate all candidates without contraindications be considered for a trial of steroids and evaluated for responsiveness with a 4- or 7-day Lille Score. Continued development and refinement of prognostic scores that can identify patients at greatest risk of death despite steroids or other future therapies need to be developed to identify patients that would obtain the greatest benefit or those where therapy is futile.

## Figures and Tables

**Figure 1 jcm-10-02756-f001:**
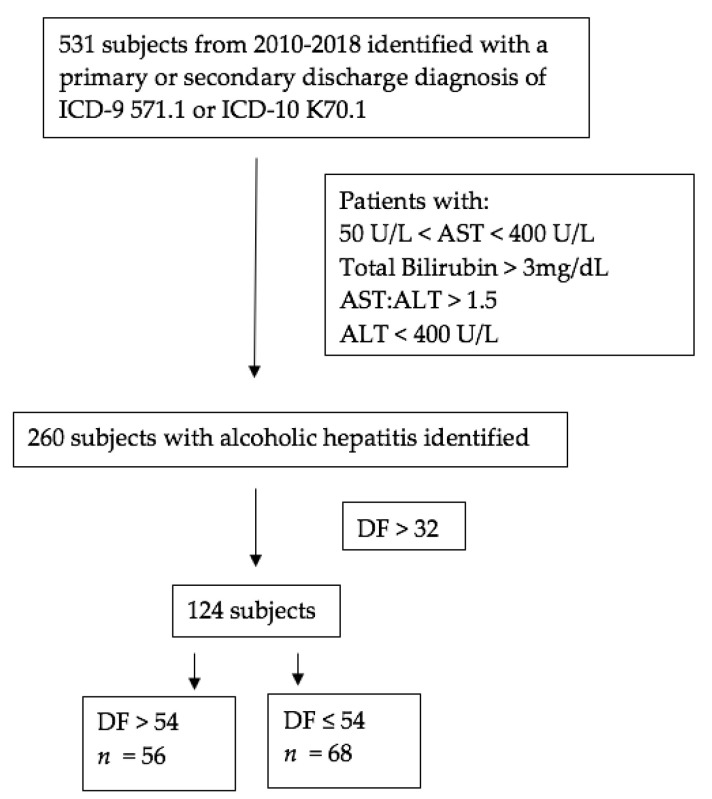
Study flow diagram.

**Figure 2 jcm-10-02756-f002:**
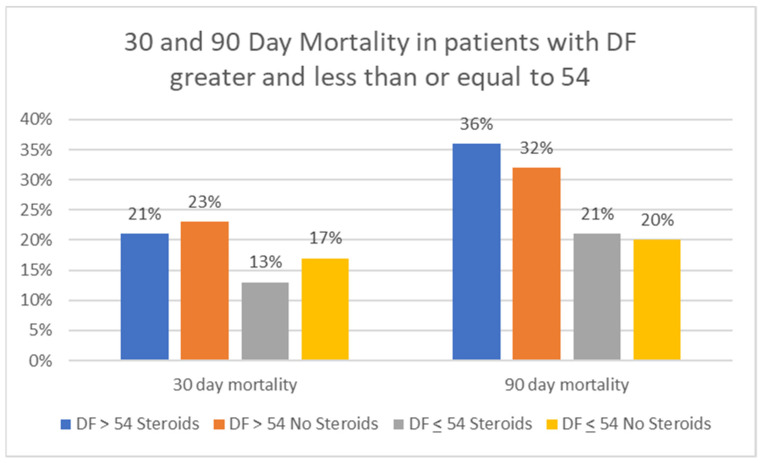
30- and 90-day mortality comparing those who did and did not receive steroids as well as DF > and < 54.

**Table 1 jcm-10-02756-t001:** Baseline Characteristics of Patients Who Received Steroids with DF > 32.

	Total *n* = 52	DF 33–54 *n* = 25	DF 55+ *n* = 27	*p*-Value
Age, years (IQR)	47 (36, 52)	47 (37, 52)	45 (34, 51)	0.36
Female (%)	15 (29)	7 (28)	8 (30)	0.90
Race				
Black (%)	8 (15)	4 (16)	4 (15)	0.85
White (%)	43 (83)	20 (80)	23 (85)	
Ethnicity				
Declined (%)	3 (6)	1 (4)	2 (7)	0.062
Hispanic or Latino (%)	4 (8)	4 (16)	0 (0)	
Non-Hispanic or Latino (%)	45 (87)	20 (80)	25 (93)	
WBC Count 10^3^/μL (IQR)	9.8 (7.5, 14.9)	9.2 (6.1, 14.7)	10.6 (8.1, 15.2)	0.28
Platelets 10^3^/μL (IQR)	127 (69, 182)	133 (98, 206)	123 (65, 157)	0.14
Sodium mmol/L (IQR)	130 (128, 134)	131 (129, 135)	129 (125, 132)	0.02
Creatinine mg/dL (IQR)	1.0 (1.0,1.3)	1.0 (1.0, 1.0)	1.0 (1.0, 1.9)	0.01
Prothrombin Time seconds (IQR)	21.0 (19.1, 22.4)	19.2 (17.7, 20.0)	22.2 (21.7, 25.0)	<0.001
INR	1.9 (1.7, 2.1)	1.6 (1.5, 1.7)	2.0 (1.9, 2.2)	<0.001
Bilirubin Day 0 mg/dL (IQR)	20.8 (14.1, 25.5)	19.9 (11.0, 21.8)	22.2 (15.8, 27.7)	0.06
Lille Score	0.69 (0.26, 0.96)	0.42 (0.23, 0.79)	0.95 (0.55, 0.99)	0.11
AST	164 (117, 239)	136 (91, 213)	171 (137, 274)	0.14
Albumin g/dL (IQR)	2.2 (1.9, 2.5)	2.3 (2.1, 2.5)	2.0 (1.8, 2.4)	0.045
Presence of GI Hemorrhage (%)	5 (10)	3 (12)	2 (7)	0.66
Infection	13 (25)	1 (4)	12 (44)	<0.001
Median MELD				
20–29 (%)	41 (79)	24 (96)	17 (63)	<0.001
30+ (%)	10 (19)	0 (0)	10 (37)	
Median MELD-Na				<0.001
20–29 (%)	36 (69)	25 (100)	11 (41)	<0.001
30+ (%)	16 (31)	0 (0)	16 (59)	

**Table 2 jcm-10-02756-t002:** Baseline Characteristics of Patients with DF > 54.

	Total *n* = 56	No Steroids *n* = 29	Steroids *n* = 27	*p*-Value
Age, years (IQR)	46 (37, 51)	46 (40, 50)	45 (34, 51)	0.27
Female (%)	19 (34)	11 (38)	8 (30)	0.51
Race				0.67
Black (%)	6 (11)	2 (7)	4 (15)	0.67
White (%)	48 (89)	25 (93)	23 (85)	
Ethnicity				
Declined (%)	5 (9)	3 (10)	2 (7)	0.67
Hispanic or Latino (%)	2 (4)	2 (7)	0 (0)	
Non-Hispanic or Latino (%)	49 (87)	24 (83)	25 (93)	
WBC Count 10^3^/μL (IQR)	10.6 (7.5,14.8)	10.6 (7.3, 13.5)	10.6 (8.1, 15.2)	0.79
Platelets 10^3^/μL (IQR)	125 (79,157)	128 (79, 162)	123 (65, 157)	0.36
Sodium mmol/L (IQR)	131 (128, 134)	132 (130, 134)	129 (125, 132)	0.001
Creatinine mg/dL (IQR)	1.0 (1.0, 2.0)	1.0 (1.0, 2.1)	1.0 (1.0, 1.9)	0.32
Prothrombin Time seconds (IQR)	23.8 (21.6, 27.3)	25.5 (21.4, 27.7)	22.2 (21.7, 25.0)	0.28
INR	2.2 (1.9,2.6)	2.4 (2.1, 2.6)	2.0 (1.9, 2.2)	0.18
Bilirubin Day 0 mg/dL (IQR)	22.2 (14.9,27.6)	21.9 (14.0, 27.5)	22.2 (15.8, 27.7)	0.43
AST	166 (123,209)	145 (111, 199)	171 (137, 274)	0.03
Albumin g/dL (IQR)	2.0 (1.8,2.3)	2.0 (1.8, 2.3)	2.0 (1.8, 2.4)	0.97
Presence of GI Hemorrhage (%)	3 (5)	1 (3)	2 (7)	0.60
Infection	15 (27)	3 (10)	12 (44)	0.004
Median MELD				
20–29 (%)	35 (62)	18 (62)	17 (63)	0.94
30+ (%)	21 (38)	11 (38)	10 (37)	
Median MELD-Na				
20–29 (%)	25 (45)	14 (48)	11 (41)	0.57
30+ (%)	31 (55)	15 (52)	16 (59)	

**Table 3 jcm-10-02756-t003:** Logistic Regression of 30- and 90- day mortality in patients with DF > 32.

**Logistic Regression of 30-Day Mortality among Patients with DF > 32**		
**Variable**	**Odds Ratio**	**95% CI (*p*-Value)**
DF > 54 vs. DF 32–54	1.67	0.65–4.32 (0.29)
Creatinine (0.3 unit increase)	1.19	1.04–1.37 (0.01)
Albumin	1.16	0.37–3.70 (0.80)
Steroid Treatment	1.01	0.39–2.61 (0.98)
Controlling for GI bleeding, age, and CCI		
**Logistic Regression of 90-Day Mortality among Patients with DF > 32**		
**Variable**	**Odds Ratio**	**95% CI (*p*-Value)**
DF > 54 vs. DF 32–54	2.05	0.88–4.79 (0.10)
Creatinine (0.3 unit increase)	1.13	1.00–1.28 (0.06)
Albumin	1.15	0.42–3.20 (0.78)
Steroid Treatment	1.46	0.64–3.33 (0.37)
Controlling for GI bleeding, age, and CCI		

**Table 4 jcm-10-02756-t004:** Logistic Regression of the Relationships between 30- and 90-Day Mortality and DF in Patients Treated with Steroids.

Outcome	Odds Ratio	95% CI (*p*-Value)
30-Day Mortality *	0.98	0.9–1.03 (0.42)
90-Day Mortality *	0.98	0.94–1.02 (0.33)
30-Day Mortality **	0.98	0.94–1.03 (0.40)
90-Day Mortality **	0.98	0.94–1.02 (0.23)

* Controlling for age and creatinine. ** Controlling for CCI and creatinine.

## Data Availability

The data presented in this study are available on request from the corresponding author. The data are not publicly available due to privacy.

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
