# Peer review of "Is There a Futility Discriminant Function Score for Alcoholic Hepatitis?"

_jcm, 2021, doi:10.3390/jcm10132756_

Round 1

Reviewer 1 Report

The scientific topic is intereting. However the biggest bias of the study is his underpowerful sample size patients. it's not surprising that the authors didn't found significant difference between groups.

Reviewer 2 Report

It would be very useful in this study to find that steroid administration does not result in 100% mortality in patients with DF>55.

However, there was no significant difference between the treated group and the non-treated group, which were 21% and 23%, respectively, p = 0.92. There is no significant difference in 90-day survival, and steroid administration has a higher mortality rate. Therefore, the usefulness of steroids remains questionable.

Twenty-seven DF54 and above patients received steroids.

I would like to know why steroids were administered to Despite reports of high mortality.

I would like to know P Value in Logistic Regression in Tables 3 and 4.

Reviewer 3 Report

In the manuscript entitled “Is there a futility discriminant function score for alcoholic hepatitis?”, the authors tried to determine if there was a DF score at which giving steroids is futile in patients with alcoholic hepatitis or not. While the findings of this study are of interest, the current study is lacking the cutting edge to be accepted for publication because of some miscalculations in the main text and figures as well as tables. To overcome this limitation, authors are recommended to modify the manuscript according to the comments which are mentioned below.

Certainly, this study is interesting in that it efficiently and systematically extracted and analyzed a large number of cases, and also in that it was able to diminish various types of bias. However, apparent discrepancies can be identified in several places regarding the number of cases listed in the abstract, Figure 1, and Tables 1 & 2. Therefore, we cannot dispel the concern that the credibility of the results of these analyses may be fundamentally compromised. It is regrettable, but we would like the authors to carefully check and correct these points again before attempting to submit the manuscript to another journal.

Reviewer 4 Report

In this manuscript Lamm et al. investigate mortality rate of alcoholic hepatitis in regard to DF score and steroid treatment. They conclude that steroids are not futile in patients with a DF score greater than 54.

The authors present a clear manuscript and I have no concerns. However, I don’t quite understand why the authors state that steroids are not futile in patients with a DF score greater than 54, as there is no significant difference in survival between the groups (treated/untreated)? Maybe authors want to explain this better. Besides this I just have a few comments that could maybe be addressed by the author if appropriate:

Specific comments:

  • I would lower the scale in the diagram in figure 2 to make the differences more visible. A scale up to 40% seems sufficient for the data and makes it easier to compare the different groups. And maybe authors could add statistics to the graph.
  • In the first paragraph in the patient and methods section the last sentence seems to be missing the age parameter. “We excluded patients less than age.”

Round 2

Reviewer 2 Report

no comment

Reviewer 3 Report

Authors have successfully corrected their mistakes.